# Combined Use of Fatty Acid Profiles and Elemental Fingerprints to Trace the Geographic Origin of Live Baits for Sports Fishing: The Solitary Tube Worm (*Diopatra neapolitana*, Annelida, Onuphidae) as a Case Study

**DOI:** 10.3390/ani14091361

**Published:** 2024-04-30

**Authors:** Fernando Ricardo, Marta Lobão Lopes, Renato Mamede, M. Rosário Domingues, Eduardo Ferreira da Silva, Carla Patinha, Ricardo Calado

**Affiliations:** 1Laboratório para a Inovação e Sustentabilidade dos Recursos Biológicos Marinhos (ECOMARE), Centro de Estudos do Ambiente e do Mar (CESAM), Departamento de Biologia, Universidade de Aveiro, Campus Universitário de Santiago, 3810-193 Aveiro, Portugal; martalopes@ua.pt (M.L.L.); renatomamede@ua.pt (R.M.); 2Centro de Estudos do Ambiento e do Mar (CESAM), Departamento de Química, Universidade de Aveiro, Campus Universitário de Santiago, 3810-193 Aveiro, Portugal; mrd@ua.pt; 3Laboratório Associado para a Química Verde (LAQV-REQUIMTE), Departamento de Química, Universidade de Aveiro, Campus Universitário de Santiago, 3810-193 Aveiro, Portugal; 4Geobiosciências, Geoengenheiria e Geotecnologias (GEOBIOTEC), Departamento de Geociências, Universidade de Aveiro, Campus Universitário de Santiago, 3810-193 Aveiro, Portugal; eafsilva@ua.pt (E.F.d.S.); cpatinha@ua.pt (C.P.)

**Keywords:** bait fishing, harvesting, polychaetes, resource management, Ria de Aveiro, sustainability

## Abstract

**Simple Summary:**

The overexploitation of the bristle worm *Diopatra neapolitana* in Ria de Aveiro, a coastal lagoon in mainland Portugal, has led to a generalized decline of its local populations, as it is commonly used as live bait for sports fishing. Several management actions have been put forward to reduce the impact of its harvesting, although illegal poaching still threatens the sustainable use of this marine resource. In an attempt to verify if *D. neapolitana* was sourced from no-take zones or if it was indeed collected from the place of origin claimed by live bait traders, this study evaluated if the geographic origin of *D. neapolitana* could be correctly assigned using a combination of fatty acid profiles and elemental fingerprints of its whole body and jaws, respectively. Results showed that both fatty acid profiles and elemental fingerprints differ significantly among locations, making it possible to discriminate the geographic origin of *D. neapolitana*. This discrimination achieves even higher accuracy when combining these two natural barcodes than when employing each one of them individually. The present work can, therefore, contribute to the enforcement of management plans for the sustainable use of this commercially important marine resource.

**Abstract:**

*Diopatra neapolitana* Delle Chiaje, 1841 (Annelida, Onuphidae) is one of the most exploited polychaete species in European waters, particularly in Ria de Aveiro, a coastal lagoon in mainland Portugal, where the overexploitation of this resource has led to a generalized decline of local populations. In an attempt to reduce the impact of harvesting, several management actions were implemented, but illegal poaching still fuels a parallel economy that threatens the sustainable use of this marine resource. The present study evaluated the combination of fatty acid profiles and elemental fingerprints of the whole body and jaws, respectively, of *D. neapolitana* collected from four harvesting locations within Ria de Aveiro in order to determine if their geographic origin could be correctly assigned post-harvesting. Results showed that both fatty acid profiles and elemental fingerprints differ significantly among locations, discriminating the geographic origin with higher accuracy when combining these two natural barcodes than when employing each individually. The present work can, therefore, contribute to the implementation of an effective management plan for the sustainable use of this marine resource, making it possible to detect if *D. neapolitana* was sourced from no-take zones and if it was collected from the place of origin claimed by live bait traders.

## 1. Introduction

The collection of polychaetes in subtidal and intertidal mudflat habitats is a widespread activity worldwide, playing a key role in coastal economies [1,2,3]. Live polychaetes are used for multiple purposes, from maturation diets for crustacean and finfish broodstock in aquaculture facilities [4] to performing bioremediation processes on marine fish farm effluents [5,6] and as live baits for recreational and commercial fishing [2]. Several polychaetes species are exploited for these purposes, such as *Arenicola marina*, *Atilla virens*, *Diopatra neapolitana*, *Glycera dibranchiata*, *Halla parthenopeia*, *Hediste diversicolor*, *Marphysa sanguinea*, *Namalycastis rhodochorde*, *Ophelia neglecta*, *Perinereis linea*, *Scoletoma impatiens*, and *Sigalion squamosus* [1,2]. Over the past decades, *D. neapolitana* has been one of the main polychaete species harvested in Ria de Aveiro, a coastal lagoon in mainland Portugal, as well as in other estuaries along the Portuguese coastline (e.g., Tagus estuary, Sado estuary, and the coastal lagoon Ria Formosa) [2,7,8,9,10]. The reproductive biology of this species is relatively unknown [7]. *D. neapolitana* is a broadcast spawner with free-swimming larvae. In Ria de Aveiro, the main reproduction peak occurs from May to August, and the male–female sex ratio is about 1:1 throughout the year [7]. This species has a regenerative capacity, being able to survive when a few anterior chaetigers are removed, mainly by predation. However, when *D. neapolitana* is harvested, usually more than 20 chaetiger are harvested, compromising the survival of the posterior part of the specimen that remains in the burrow it inhabits [7]. The overexploitation of this resource has prompted a decline in local populations [1], and in an attempt to reduce the impact of the harvesting activity, several measures have subsequently been implemented. According to Portuguese legislation (Portaria n° 1228/2010), bait gatherers can only operate with a personal license and are only allowed to work using hand gathering or restricted gear. A more recent Ordinance was published in January 2014 (Portaria n° 14/2014) in an attempt to define the maximum daily catch limit. The daily limit is assumed to reflect the maximum sustainable yield (MSY) of digging, ensuring a sustainable income for diggers.

According to this Ordinance, the daily catch limit for annelids should be 0.5 L day^−1^ per digger, excluding the tube [11]. In 1999, a commercial bait harvesting value of around 200 million euros was estimated for Europe, but the existing gaps in the supervision and regulation of this activity could result in an underestimation of the real value [12]. In 2001/2002, the harvesting of *D. neapolitana* was quantified for the first time in Ria de Aveiro, with an estimated annual catch volume of around 45 tons and a commercial value of approximately 350 thousand euros [13]. Nevertheless, most commercial diggers, at a national and international level, do not have valid licenses nor properly report their harvesting; as such, they contribute to fostering a parallel economy by not declaring their sales for tax purposes and impair any management plan ruling this activity to be successfully implemented [12,14]. The commercial potential of live fishing bait is so high that several attempts were made to intensively culture *D. neapolitana* [15,16]. Nonetheless, several constraints still hamper this approach, and the harvesting of specimens from wild populations remains the sole supply source for this highly priced polychaete species.

The effects of bait harvesting on species density and population structure have a direct impact on the community (neglecting non-target species) and, consequently, on ecosystem functioning and processes [17,18]. The polychaete population’s response to harvesting is influenced by the nature and extent of bait-digging pressure and by the demographic characteristics of the population being exploited [14]. The current status of this species and the sustainability of its harvest are issues of growing concern [11]. As such, reliably tracing the geographic origin of polychaetes harvested from the wild could be paramount for a sustainable management plan for bait harvesting that could foresee no take periods and areas, as well as expose illegal poaching activities by less scrupulous bait diggers.

At present, several tools have been applied to confirm the geographic origin of marine organisms [19,20]. Fatty acid (FA) profiles recorded in soft tissues (e.g., adductor muscle of bivalves) have been successfully used for this purpose in common cockles (*Cerastoderma edule*) [21,22] and manila clams (*Ruditapes decussatus*) [23]. The particular physicochemical conditions of each ecosystem shape the FA composition of marine organisms in the sense that salinity and temperature are known to modulate the structure, fluidity, and, thus, the composition of cell membranes [24]. Higher salinity fluctuations and/or lower water temperatures promote a decrease in the levels of saturated FA (SFA) and an increase in the concentration of polyunsaturated FA (PUFA), responsible for the stabilization of the bilayer structure [24]. Elemental fingerprints (EFs) use the elemental profile recorded in hard biogenic structures, such as shells of common cockles (*C. edule*) [25,26], otoliths of California halibut (*Paralichthys californicus*) and Garibaldi (*Hypsypops rubicundus*) [27], and bony plates of Long-Snouted Seahorse (*Hippocampus guttulatus*) [28]. Considering that elements are influenced by the environmental and chemical features of each ecosystem [29] and that these mineral structures grow throughout the year, EFs have already been successfully used to discriminate specimens originating from geographically close locations [30,31,32].

The combination of two different traceability tools can be more effective in confirming the geographic origin of marine organisms, as already confirmed by Zhang et al. [33] and Perez et al. [34], who showed that the use of stable isotope ratios in combination with FA profiles could successfully discriminate scallop species (*Patinopecten yessoensis*, *Chlamys farreri*, and *Argopecten irradians*) and warty venus (*Venus vecurrosa*), respectively, from different geographic locations. This combination of tools was also able to trace the geographic origin and seasonality of the whitemouth croaker (*Micropogonias furnieri*) [35]. Moreover, Matos et al. [36], using stable isotopes and elemental fingerprints, effectively traced the geographic origin of eastern oysters (*Crassostrea virginica*).

In an attempt to contribute to better management of *D. neapolitana* stocks, the present study tested, for the first time, if the combination of FA profiles of the whole polychaete body and the EF of its jaws differs between specimens originating from different locations in Ria de Aveiro, a coastal lagoon in mainland Portugal, where the capture of this polychaete being used as live bait for sports fishing is an important economic and social activity.

## 2. Materials and Methods

### 2.1. Sample Collection

A total of forty adult specimens of *D. neapolitana* (total body length ranging from 314 mm to 584 mm) were randomly collected during low tide during the Spring of 2015 using a shovel, mimicking the method used by professional bait collectors in Espinheiro (E), Ílhavo (I), and Mira (M1 and M2) channels located in Ria de Aveiro, mainland Portugal (Figure 1). This coastal lagoon is one of the most important locations for the harvesting of *D. neapolitana* for recreational and commercial fishing on mainland Portugal [2,37]. Ten specimens of *D. neapolitana* were sourced from each location described above, stored in aseptic plastic boxes, and kept refrigerated during sampling and transport to the laboratory. Upon arrival at the laboratory, specimens were left to depurate for 24 h in containers with artificial seawater (prepared by mixing Tropic Marin Pro Reef salt (Tropic Marin, Wartenberg, Germany) and freshwater purified by a reverse osmosis unit). All specimens were split up into two sub-groups: whole body for FA analysis and jaw for elemental analysis (4 locations × 2 methods × 10 replicates = 80 samples). The jaws of each specimen were dissected using a scalpel with ceramic-coated blades. Subsequently, the whole body and jaws were individually homogenized using a mortar grinder (RM 200, Retsch, Hann, Germany) and stored at −80 °C until further analysis.

### 2.2. Fatty Acid Analysis

Methyl esters of fatty acids (FAME) of *D. neapolitana* (~50 mg) were prepared following the method described by Aued-Pimentel et al. [38] (through transmethylation of FA using a mixture of methanolic solution KOH (2 M) and saturated NaCl). The resulting FAME were analyzed in a QP2010 Ultra Shimadzu gas chromatography–mass spectrometry, equipped with an auto-sampler a DB-FFAP column with 30 m length, 0.32 mm internal diameter, and 0.25 µm film thickness (J&W Scientific, Folsom, CA, USA). The column was initially programmed to 80 °C, increasing 25 °C min^−1^ until 160 °C, 2 °C min^−1^ from 160 to 220 °C, and 30 °C min^−1^ from 220 to 250 °C, using helium as the carrier gas, at a flow of 1.8 mL min^−1^. FAME identification was accomplished through comparison of retention times with those mixed FAME standards (C4–C24, Supelco 37 Component Fame Mix) and by comparison of the mass spectrum of each relative to standard spectra from the library “AOCS Lipid Library” (http://lipidlibrary.aocs.org/ (accessed on 5 January 2024)).

### 2.3. ICP-MS

The jaws of *D. neapolitana* (~50 mg) were weighed in digestion tubes (DigiTUBEs). Digestion tubes were soaked with high-purity concentrated HNO_3_, HCl (37%), and H_2_O_2_ (30% *w*/*v*) on a digestion block (DigiPrep, SCP Science, Baie-d’Urfé, QC, Canada) at 85 °C for over 15 min. Subsequently, solutions were diluted with Milli-Q (Millipore) water to a final HNO_3_ concentration of 1–2% to reduce acid concentration and prevent damage to the equipment. Finally, total concentrations of aluminum (Al), barium (Ba), calcium (Ca), cerium (Ce), cobalt (Co), iron (Fe), potassium (K), lanthanum (La), magnesium (Mg), manganese (Mn), sodium (Na), nickel (Ni), phosphorus (P), strontium (Sr), and yttrium (Y) were analyzed using an Agilent 7700 ICP-MS equipped with an octopole reaction system (ORS) collision/reaction cell technology to minimize spectral interferences. Blanks and certified reference materials, BCS-CRM-513 (SGT Limestone 1; LGC Standards, UK), were treated using the same method.

### 2.4. Data and Statistical Analysis

#### 2.4.1. Analysis of Fatty Acid Profiles

The FA profiles were represented by the relative abundance of the total pool of FA per replicate for each location. The FA were grouped by the following classes: saturated FA (SFA), monounsaturated FA (MUFA), and polyunsaturated FA (PUFA). A Boruta analysis [39] was performed to select the most relevant FA to discriminate specimens originating from different sampling locations. The outcome grouped the variables into three categories: tentative variables (not enough to accept or reject), confirmed variables, and rejected variables. For the tentative variables, the “TentativeRoughFix” function was applied. Statistical differences (*p* < 0.05) in FA profiles among locations were tested using the vegan adonis function for permutational multivariate analysis of variance (PERMANOVA) [40] using Euclidean distances. To test differences among locations for each FA individually, a one-way analysis of variance (ANOVA) was performed. A Radom Forest (RF) classification [41] was used to evaluate the possibility of successfully discriminating the geographic origin of specimens through the FA profile. To test for normality and the assumption of homogenous variances, the Shapiro.test and bartlett.test functions were used, respectively, and data were transformed (log x+1). All statistical analyses were performed in R [42]. The PERMANOVA and ANOVA were performed using the package “vegan,” while Boruta analysis and RF were performed using the package “Random Forest”.

#### 2.4.2. Analysis of Elemental Fingerprint

The concentration of elements present in the jaws of *D. neapolitana* was expressed as a ratio relative to Ca (mg/mg) in order to minimize mass effects [25,28,43]. For a better understanding of our results, a Boruta analysis using the “TentativeRoughFix” function [39] (see above for details) was used to determine the most important elements to discriminate between different sampling locations. A resemblance matrix using the ratio of each element per sample was prepared using Euclidean distance with the vegdist function after performing a scale transformation. A PERMANOVA was performed to detect significant differences in the EF of jaws from *D. neapolitana* originating from different locations. A one-way ANOVA was used to assess differences among locations for each individual element after confirming normality with the Shapiro test and homogeneity of variance with the Bartlett test. An RF classification was performed to evaluate the potential use of elements present in the jaws of this polychaete to discriminate between different locations. All statistical analyses were performed in R [43] (see above for details about R packages).

#### 2.4.3. Combination of Fatty Acid Profiles and Elemental Fingerprint

In order to increase the number of predictor variables, a combination of FA profiles and EF was used. This combination underwent the application of the ‘TentativeRoughFix’ function within a Boruta analysis [39] (see details in Section 2.4.1) to select the best subset of variables that may explain potential differences in specimens of *D. neapolitana* originating from different sampling locations [44]. A PERMANOVA was performed to assess differences among locations, and an RF classification was applied to test if the combination of FA and EF could be used to predict the geographic origin from which each polychaete was collected. All data were scale transformed and used to produce a matrix using Euclidean distances with the vegdist function in the vegan package. All statistical analyses were performed in R [42] (see above for details about R packages).

## 3. Results

### 3.1. Fatty Acid Profiles

The average FA profiles recorded for the whole body of *D. neapolitana* at different locations are presented in Appendix A. A total of twenty-four FAs were identified, comprising eight SFA, six MUFA, and ten PUFA. SFA comprised 40% and 50% (locations M2 and I, respectively; Appendix A) of all FA identified in the whole body of *D. neapolitana* from different locations. The main SFAs were palmitic acid (16:0) and stearic acid (18:0), corresponding to more than 76% of all SFAs. MUFA represented 14% and 20% (locations M1 and E, respectively; Appendix A) of all FA. Dominant MUFAs were vaccenic (18:1n-7) and eicosenoic (20:1n-9) acids, representing over 50% of all MUFAs. PUFA comprised 32% and 43% (locations E and M2, respectively; Appendix A) of all FA. The most abundant PUFA were eicosapentaenoic (20:5n-3; EPA) and docosadienoic (22:2n-9; DHA) acids, representing over 42% of all PUFA.

The Boruta analysis performed showed that the FA 18:1n-7, 20:0, 18:2n-6, 16:0, 15:0, 22:4n-6, 17:0, 16:1n-7, 20:4n-6, 18:3n-3, 18:0, 20:1n-11, and 20:5n-3 (Figure 2a) were the ones that most contributed to the differences recorded among locations. The PERMANOVA performed revealed the existence of significant differences among locations, apart from those within the Mira Channel (M1 and M2) (*p* < 0.05; Appendix A). Considering each FA individually, FA 22:1n-11 and FA 20:5n-3 did not display any significant difference among locations (Appendix A). At the same time, no significant differences in FA 20:0 were only recorded between M1 and I (Appendix A). Comparatively to the other locations, specimens from location E recorded significantly higher levels of 15:0, 16:0, 17:0, 18:1n-7, 18:2n-6, and 18:3n-3 (Appendix A). Specimens from location M2 presented the highest levels of 16:1n-7, 20:4n-6, and 22:4n-6, being significantly different from other locations (Appendix A). Location I presented higher levels of FA 18:0, with significant differences between this location and the others; the sole exception was with M1.

The RF classification showed an overall accuracy of 77.5% (Table 1 and Figure 3a). Specimens from location E exhibited the highest percentage of correct classification (100%; UA—100%). Specimens of *D. neapolitana* originating from Mira Channel (M1 and M2) registered two and three misclassified replicates, resulting in a correct classification of 70 (UC and 80%). Most misclassifications were associated with specimens sourced from location I, with only 60% of correct classifications (Figure 3a and Table 1).

### 3.2. Elemental Fingerprints

The concentration of the fourteen elements recorded on the EF determined for the jaws of *D. neapolitana* in the different locations of Ria de Aveiro is shown in Appendix A. The Boruta analysis revealed that Ba, Mn, K, Na, Ni, and Y were the elements that most contributed to the differences recorded among locations (Figure 2b). The PERMANOVA revealed significant differences among locations, with the exception of specimens collected in locations E vs. I and M1 vs. I (Appendix A). Concerning individual elements, all specimens collected in the locations surveyed in this study displayed a similar Y ratio for this element. The highest concentration of Ba was registered in location M2, with significant differences occurring between locations M2 vs. E, M2 vs. I, and M1 vs. E (Appendix A). The jaws of *D. neapolitana* presented the highest levels of Mn in location M2, with significant differences being recorded between this location and location E. Specimens from M1 registered significantly higher concentrations of Na, compared with locations E and I (Appendix A). Concerning Ni, the highest concentrations were recorded in specimens from locations M2 and I, whereas K was more concentrated in specimens from location M1, with no significant differences being recorded among locations for both of these elements (Appendix A).

The RF classification revealed an accuracy of 72.5% when using EF to assign *D. neapolitana* to its geographical origin. Specimens from location I showed the highest percentage of correct allocation (90%; one replicate was misclassified), followed by those originating from location M1, with an accuracy of 80% (two replicates were misclassified). Most misclassifications were associated with *D. neapolitana* collected in locations E and M2, with 40% of collected specimens being erroneously assigned to other locations (Figure 3b and Table 1).

### 3.3. Elemental Fingerprints

The Boruta analysis showed that the most relevant combination of predictive variables when simultaneously considering FA profiles and EF were Ba, 18:1n-7, 18:2n-6, 16:0, 20:0, 15:0, 17:0, 22:4n-6, 18:3n-3, 16:1n-7, 20:4n-6, 18:0, 22:1n-11, Ni, 20:5n-3, and Na (Figure 2c). The PERMANOVA revealed the existence of significant differences among all locations (Appendix A). The RF classification resulted in an overall accuracy of 87.5% (Figure 3c and Table 1). Location E had the highest percentage of correct allocation (100%), whereas single replicates from locations I and two replicates from locations M1 and M2 were misclassified, resulting in an overall correct allocation of 90 and 80%, respectively.

## 4. Discussion

The use of FA profiles of soft tissues and EF of mineral structures in marine species has been optimized to put forward faster and more accurate methods of analysis that can also reduce potential environmental impacts (associated with the residues they generate) that allow to best discriminate the geographic origin of these organisms, namely those that feature an important commercial value [23,28,44,45,46,47,48,49,50]. Most of the available studies to date using these methods, either when they are applied individually or combining more than one approach (e.g., FA profiles and stable isotopes [34], elemental fingerprints and stable isotopes [36]), are mostly focused on food safety issues rather than the implementation of effective management plans for endogenous marine resources that may be vulnerable to poaching. The present study showed, for the first time, that the combination of biogeochemical tools (FA profiles and EF) using the whole body and the jaws of *D. neapolitana* (respectively) can successfully be used to confirm the geographic origin of bait digging with a high accuracy level (Table 1). It is, therefore, legitimate to say that these natural barcodes can be successfully used towards the implementation of more effective fishery management plans, allowing the enforcement of no-take zones.

The FA profiles displayed by the whole body of *D. neapolitana* revealed that the most dominant FA was 16:0, followed by the PUFA 20:5n-3. In general, this trend was similar to that found for other polychaetes by Fernandes et al. [50] for *Hediste diversicolor* and by Jerónimo et al. [51] for *H. diversicolor*, *D. neapolitana, Sabella* cf. *pavonina*, and *Terebella lapidaria*, all collected in the same location in Ria de Aveiro. The same was found for *Alvinella pompejana* [52], *Arenicola marina* [53], *Nepthys hombergii* and *Lanice conchilega* [54] sampled in other locations.

The FA belonging to the SFA and PUFA classes were responsible for most of the differences recorded among locations (*p* < 0.05; Appendix A). These dissimilarities in FA profiles of polychaetes among locations were likely associated with a differential physiological response to changes in environmental conditions (e.g., salinity and temperature) that shape the environment in their sampling locations [22,50,55,56]. At higher temperatures, the reorganization of the membrane structure is needed to maintain membrane fluidity and homeostasis, leading to an increase in FA saturation or the prevalence of shorter-chain FA [57]. Salinity is responsible for changes in FA profiles involved in the osmoregulation process, inducing changes in membrane permeability [58]. Higher salinities, like those registered at M1, I, and E, are associated with a decrease in PUFA to reduce membrane permeability [59].

The biogenic carbonate-hard parts of marine species, such as shells, otoliths, plates, fish scales, and fish bones, incorporate and retain elements from the surrounding environment throughout their lifetimes [28,48,60,61,62,63]. It is important to note that this study represents the first dataset reporting the EF of polychaete jaws. Similar to other mineral structures (e.g., *Cerastoderma edule* shells [26,64]), the levels of elements recorded in the aragonitic jaws of *D. neapolitana* [65] differed among locations within Ria de Aveiro. These discrepancies are associated with both local physical conditions and the availability of elements in the environment. Considering that location M2 is more upstream than M1 (within the same channel and closer to the inlet), hydrological conditions differ, which results in a substantial enrichment of Ba and Mn upstream due to freshwater inputs and nutrient runoff, as suggested by Ricardo et al. [26]. In fact, the presence of high levels of Ba and Mn in location M2 had already been previously reported by these authors in *C. edule* shells [26]. The highest concentrations of K and Na observed in M1 could be related to its geographical proximity to the inlet of the coastal lagoon, suggesting a potential association with the physicochemical properties of oceanic seawater (e.g., temperature and salinity [66,67]). Locations I and M2 registered the highest levels of Ni in *D. neapolitana* jaws. This trend was previously reported for the body of *D. neapolitana* by Pires et al. [68], exactly in those same locations. High levels of Ni could be associated with anthropogenic impacts [69,70], such as the presence of an important shipyard, commercial harbor, and industrial activities in location I [71], along with agriculture runoff in location M2 [72]. Similarities in the EF of *D. neapolitana* jaws between locations E and I and locations M1 and I (Appendix A) were likely associated with the similar environmental conditions experienced by the specimens sampled in these locations.

The use of Random Forest classifications based on FA profiles and EF enhanced the discrimination of geographic origin between specimens of *D. neapolitana* (Table 1). Indeed, the use of each tool individually was associated with some constraints. When FA profiles were used alone to determine the geographic origin of collected specimens, location E was well discriminated against, contrary to the other locations. When using EF, locations I and M1 were discriminated against with a high level of accuracy, contrary to locations E and M2 (Table 1). Thus, the combination of FA profiles and EF proved to be a more efficient approach to successfully allocating sampled specimens of *D. neapolitana* to their true geographic origin, thus increasing the success rate (Table 1). The combination of different fingerprinting methods (e.g., FA profiles with stable isotopes or multi-elements with stable isotopes) had already been successfully employed to discriminate the geographic origin of different species. When using FA profiles combined with stable isotope analysis, the geographic origin of sea cucumbers (*Apostichopus japonicus*) [73], jumbo squids (*Dosidicus gigas*) [74], and scallops (*Patinopecten yessoensis*, *Chlamys farreri*, and *Argopecten irradians*) [33] was determined with a high success rate of correct allocations. Employing a multi-element and stable isotope approach for different prawn species (*Penaeus indicus*, *P. merguiensis*, *P. monodon*, *P. notialis*, *P. vannamei*, *Pleoticus muelleri*, and *Pandalus borealis*) was also highly successful when aiming to allocate their geographic origin [75]. These approaches were also used to distinguish production methods (farmed vs. wild), namely by using FA profiles and stable isotopes on European eel (*Anguilla anguilla*) [76] and Atlantic salmon (*Salmo salar*) [77], as well as for prawns when using multi-elements and stable isotope analysis (see above; [74]).

## 5. Conclusions

The present study showed that the combination of FA profiles of the body of *D. neapolitana* and EF of its jaws is an accurate and reliable traceability tool that can be used to confirm the geographic origin of live specimens of this species sourced from different locations within a coastal lagoon with a resolution < 2 km. Despite being an initial study, the results here presented can play a key role in supporting the implementation of conservation/management plans for bait harvesting activities in this coastal lagoon and elsewhere, namely estuarine systems where bait harvesting is more intense and relevant from an economic, social, and ecological perspective. The use of statistical methods, such as the Boruta selection function used in this study, can also be applied in other areas of marine science, such as benthic habitat mapping [78]. The selection performed by Boruta helps to achieve more robust and accurate models, as well as to understand which explanatory variables (e.g., elements and FA) characterize each categorical response variable (e.g., location).” To improve the potential use of this tool, further insights are required on the existence of temporal variability (both seasonal and interannual), which pinpoints how frequently the predictive model needs to be calibrated to secure an accurate classification of the place of origin of the specimens being surveyed. Extending the application of this methodology to other polychaete species with commercial interest could also be important to contribute to a more sustainable use of these important marine living resources.

## Figures and Tables

**Figure 1 animals-14-01361-f001:**
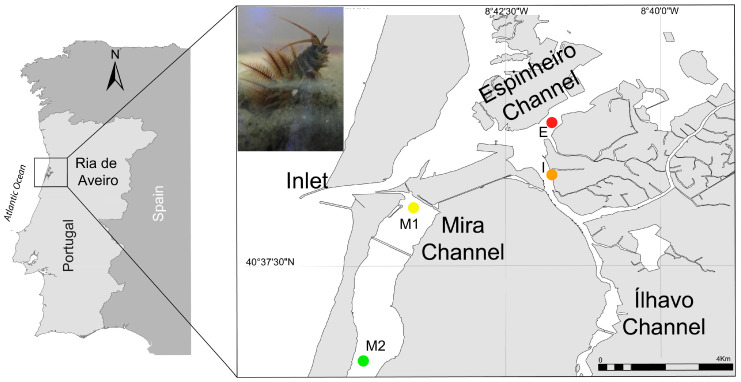
Sampling locations of *Diopatra neapolitana* in the coastal lagoon Ria de Aveiro, mainland Portugal: Espinheiro Channel (E: 40°39′48.50″ N, 8°41′45.03” W), Ílhavo Channel (I: 40°38′35.40″ N, 8°41′35.40″ W), and Mira Channel (M1: 40°38′26.30″ N, 8°43′58.90″ W and M2: 40°35′58.30″ N, 8°44′47.80″ W). Top left corner: picture of a *Diopatra neapolitana*. Photo: Fernando Ricardo, Universidade de Aveiro.

**Figure 2 animals-14-01361-f002:**
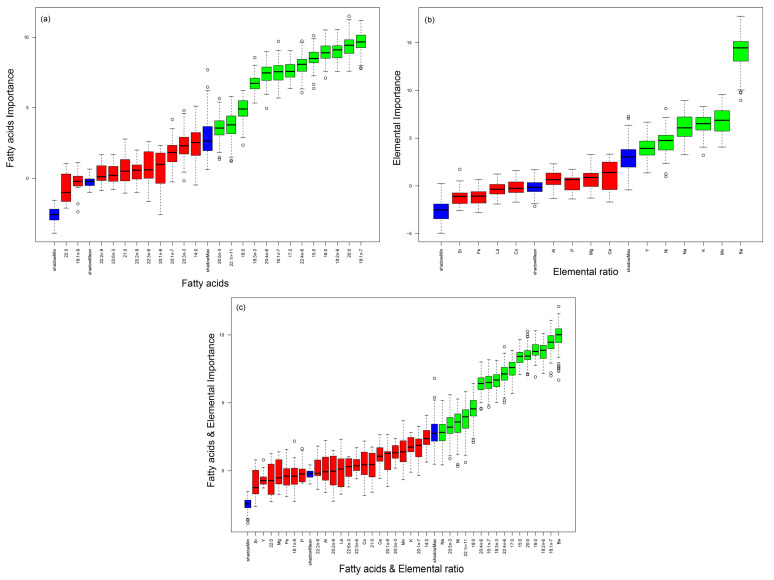
Box plots of Z-scores registered by the Boruta algorithm were used to determine the most relevant (**a**) fatty acids present on the whole body of *Diopatra neapolitana*, (**b**) elemental ratios of the jaws of these polychaetes, and (**c**) a combination of fatty acids from the whole body and the elemental ratios of the jaws of these organisms sourced from different locations within the coastal lagoon Ria de Aveiro, mainland Portugal. Shadow variables are in blue, variables confirmed as unimportant are in red, and the most relevant are in green.

**Figure 3 animals-14-01361-f003:**
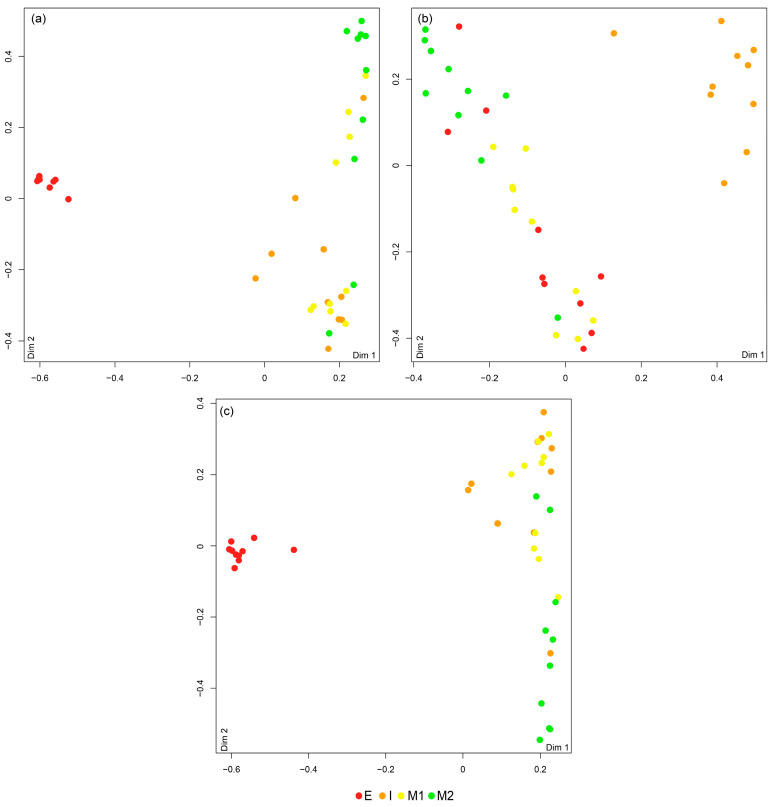
Multidimensional scaling (MDS) ordinations of proximity scores from Random Forest classifications based on (**a**) fatty acids present on the whole body of Diopatra neapolitan, (**b**) elemental ratios of the jaws of these polychaetes, and (**c**) a combination of fatty acids from the whole body and the elemental ratios of the jaws of these organisms sourced from different locations within the coastal lagoon Ria de Aveiro, mainland Portugal. Espinheiro Channel (E), Ílhavo Channel (I), and Mira Channel (M1 and M2).

**Table 1 animals-14-01361-t001:** This classification success (by location and structure(s)) of the Random Forest model when using fatty acid profiles of the whole body, b) elemental ratios of jaws, and c) a combination of fatty acid profiles of the whole body and elemental ratios of jaws of the polychaete *Diopatra neapolitana*. Espinheiro Channel (E), Ílhavo Channel (I), and Mira Channel (M1 and M2).

Structure (s) Fingerprint (s)	Original Location	Predicted Location				Total per Location	% Correct	% Correct (Location)
		E	I	M1	M2			
Whole body—FA profile	E	10	0	0	0	10	100	77.5
	I	0	6	3	1	10	60	
	M1	0	2	7	1	10	70	
	M2	0	1	1	8	10	80	
User accuracy		100	67	64	80			
Jaws—EF	E	6	1	0	3	10	60	72.5
	I	0	9	0	1	10	90	
	M1	2	0	8	0	10	80	
	M2	3	0	1	6	10	60	
User accuracy		55	90	89	60			
Whole body and Jaws—FA profile and EF	E	10	0	0	0	10	100	87.5
	I	0	9	0	1	10	90	
	M1	0	1	8	1	10	80	
	M2	0	1	1	8	10	80	
User accuracy		100	82	89	80			

## Data Availability

All raw data on the relative abundance (%) of fatty acids (FAs) and ICP-MS analysis are available as Appendix A, respectively).

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
