# Peer review of "Combined Use of Fatty Acid Profiles and Elemental Fingerprints to Trace the Geographic Origin of Live Baits for Sports Fishing: The Solitary Tube Worm (Diopatra neapolitana, Annelida, Onuphidae) as a Case Study"

_animals, 2024, doi:10.3390/ani14091361_

Round 1
Reviewer 1 Report
Comments and Suggestions for Authors
Dear author/authors,
It is my pleasure to review the manuscript (MS) with the title “Combined use of fatty acid profiles and elemental fingerprints to trace the geographic origin of live baits for sports fishing: the solitary tube worm (Diopatra neapolitana) as a case study”. In general, the manuscript contains original and valuable research information. The manuscript is well written and organized, obtaining relevant results that can be applied to contribute for implementing an effective management plan for the sustainable use of the studied resources. However, a rigorous English and references check, and some contents are necessary. The manuscript could be accepted after carefully addressing the following issues and also looking into the manuscript.

Some minor English language is required.
Author Response
Revision letter
Replies to each reviewer comment are provided bellow as RxRy (with x being the number of the reviewer and y the number of the comment by the same reviewer).
The anonymous reviewers are acknowledged, as their constructive criticism helped to improve the overall quality of the final manuscript.
Reviewer 1
R1C1: Line 72 “…0.5 L day-1…” Superscript.
R1R1: Corrected as suggested.
R1C2: Line 151 – Delete “biomass”
R1R2: Deleted as suggested.
R1C3: Line 159 “…increasing 25 °C min-1 …” Superscript.
R1R3: Corrected as suggested.
R1C4: Lines 168 and 171 “…HNO3, HCl (37%) and H2O2…” and “…HNO3 concentration
…”, respectively - Subscript.
R1R4: Despite the suggested correction appearing as ‘delete’, we interpreted it as a mistake in the subscript format. It now reads “…HNO3, HCl (37%) and H2O2…” and “…HNO3 concentration”.
R1C5: Line 194 “…statistical analyses were performed using R [40]” Please mention the package. Line 208 “…statistical analyses were performed using R [40]” Package???. Line 221 “…statistical analyses were performed using R [40]” Package???.
R1R5: As suggested, in order to provide more information about the packages used in the statistical analysis, the sentence was revised and it now reads: “All statistical analyses were performed in R [40]. The PERMANOVA and ANOVA were performed using package “vegan”, while Boruta analysis and RF were performed using package “Random Forest”. (Lines 203-205).
R1C6: I am confuse about the explanation figure 2 in the text. Please maintain a logical writing. Before explaining the figure 3 please explain figure 2. Thank you
R1R6: While we acknowledge the positive criticism by Reviewer 1, we respectfully disagree We consider that explaining Figure 2a followed by Figure 3a and then Figure 2b followed by Figure 3b, instead of Figures 2a and 2b followed by Figures 3a and 3b, makes the presentation of results is more logical and more easily understandable by the reader. However, if the Editor considers that this recommendation by Reviewer 1 is paramount to improve the overall quality of our revised manuscript the authors will address it.
R1C7: Line 383 – delete “…(see above; …).
R1R7: Corrected as suggested.
R1C8: References 36 and 38 lines (519 and 524, respectively) add “.”
R1R8: Corrected as suggested.
Reviewer 2 Report
Comments and Suggestions for Authors
My suggestions and comments on your manuscript are as follows:
1. Include the name of the family of the worked species in the title, summaries and introduction. Onuphidae.
2. Lines 19 and 34, the name of the author of the species should not be in parentheses.
3. Line 30, I guess there is an error and it is IN and not I.
4. Line 192 is missing a space and the parenthesis that follows is empty
5. Lines 227, 230 and 233 instead of hyphen between the values ​​put the word "and"
6. Lines 267 and 268, the sentence where the exception is mentioned is not explained in Discussion.
7. Line 256 in quote 65 in line 581 Anadara senilis is not italicized.
8. Line 374 includes a squid and in quote 33 the title of the work only includes bivalves, please check if another quote is missing with the information on Dosidicus gigas.
Author Response
Revision letter
Replies to each reviewer comment are provided bellow as RxRy (with x being the number of the reviewer and y the number of the comment by the same reviewer).
The anonymous reviewers are acknowledged, as their constructive criticism helped to improve the overall quality of the final manuscript.
Reviewer 2
R2C1: Include the name of the family of the worked species in the title, summaries and introduction. Onuphidae.
R2R1: Corrected as suggested.
R2C2: Lines 19 and 34, the name of the author of the species should not be in parentheses.
R2R2: Corrected as suggested. We acknowledge Reviewer 2 for flagging this mistake.
R2C3: Line 30, I guess there is an error and it is IN and not I.
R2R3: Thank you for spotting this mistake. We corrected it accordingly.
R2C4: Line 192 is missing a space and the parenthesis that follows is empty
R2R4: Thank you for spotting this mistake. We corrected it accordingly.
R2C5: Lines 227, 230 and 233 instead of hyphen between the values put the word "and"
R2R5: Corrected as suggested.
R2C6: Lines 267 and 268, the sentence where the exception is mentioned is not explained in Discussion.
R2R6: The reviewer raises a very important issue. To best clarify this topic we have now added information in our revised discussion that reads as follows: "Similarities in the EF of D. neapolitana jaws between locations E and I, and locations M1 and I (Table S2; supplementary information), were likely associated with the similarity of the environmental conditions experienced by the specimens sampled on these locations." (Lines 378-381)
R2C7: Line 256 in quote 65 in line 581 Anadara senilis is not italicized.
R2R7: Thank you for spotting this mistake. We revised it accordingly.
R2C8: Line 374 includes a squid and in quote 33 the title of the work only includes bivalves, please check if another quote is missing with the information on Dosidicus gigas.
R2R8: Thank you for spotting this omission. We have now included the missing reference in our revised version: 75 - Gong, Y.; Li, Y.; Chen, X.; Chen, L. Potential use of stable isotope and fatty acid analyses for traceability of geographic origins of jumbo squid (Dosidicus gigas). Rapid Commun. Mass Spectrom. 2018, 32, 583-589.
Reviewer 3 Report
Comments and Suggestions for Authors
The research paper is commendably written, adhering to the contemporary standards of benthic habitat studies. However, it could benefit from minor enhancements. For a few suggested modifications, please refer to the attached PDF file where you will find more comprehensive feedback.

Author Response
Revision letter
Replies to each reviewer comment are provided bellow as RxRy (with x being the number of the reviewer and y the number of the comment by the same reviewer).
The anonymous reviewers are acknowledged, as their constructive criticism helped to improve the overall quality of the final manuscript.
Reviewer 3
R3C1: Line 184 - Please supplement the proper reference for Boruta feature selection algoritm: 10.18637/jss.v036.i11
R3R1: The additional information was included as suggested.
R3C2: Line 190 - Please supplement the proper reference for Random Forest classifier: 10.1023/a:1010933404324
R3R2: The additional information was included as suggested.
R3C3: Line 194 - Please also supplement the R libraries that you used for your data analysis
R3R3: Please refer to R1R5.
R3C4: It is not advisable to leave the tentative variables in their current state. There are several strategies to enhance the results from the Boruta algorithm. For instance, you could increase the maximum number of iterations or incorporate the TentativeRoughFix algorithm into your methodology. These adjustments could potentially lead to more robust and reliable outcomes.
R3R4: To best accommodate this important remark by Reviewer 3, we have revised the “Data and statistical analysis” in section “Material and methods”. It now reads as follows: “The outcome grouped the variables into three categories: tentative variables (not enough to accept or reject), confirmed variables, and rejected variables. For the tentative variables, the “TentativeRoughFix” function was applied.” (Lines 293-295). Considering that some of the variables changed from tentative variables to confirmed variables, the statistical analyses were revised, as well as associated figures, thus making possible to slightly improve some of the outputs of our analysis.
R3C5: Table 1 - potentially punctuation error
R3R5: Thank you for spotting this mistake. We corrected it as suggested.
R3C6: In the error matrices, it would be beneficial to incorporate both the sums of predicted and reference observations. Additionally, including user’s and producer’s accuracies could provide a more comprehensive analysis. For further guidance, please refer to the following reference: 10.1016/s0034-4257(01)00295-4
R3R6: While Reviewer 3 suggestion is pertinent, we consider that sums of predicted, reference observations and producer’s accuracies are already mentioned in Table 1, although using a different terminology. Total per location corresponds to reference observations, predicted location to sums of predicted and % of correct to producer’s accuracies. User’s accuracy was added to Table 1 as well suggested by Reviewer 3. We hope this correction is enough to best accommodate the suggestion provided.
R3C7: Line 362 - It would be beneficial to state also the use of Boruta feature selection algorithm and its inplications for benthic scientific community. For further guidance, you may refer to the following benthic habitat mapping works: - 10.1016/j.ecss.2022.108053 - 10.3390/rs13234771
R3R7: To best accommodate this important suggestion by Reviewer 3, we have now added the following information in the conclusion of our revised manuscript: “The use of statistical methods, such as the Boruta selection function used in this study, can also be applied in other areas of marine science, such as benthic habitat mapping [79]. The selection performed by Boruta helps to achieve more robust and accurate models, as well as to understand which explanatory variables (e.g. elements and FA) characterize each categorical response variable (e.g. location)." (Lines 413-417).
Reviewer 4 Report
Comments and Suggestions for Authors
line 54 purposes
line 311 environmently safer please explain what that aims to
there is no effect of the metod on profiles on the environment. I believe you want to explain the effect of the knowledge on the possibility to develop environmental work for the sustained populations. Please try to reformulate to improve the understanding.
i suggest that you specify the season when catch of samples was performed at different locations , to be able to exclude season as the reason to your results..Temperature and light have effect on algal composition and thereby the food chain.
I t is also needed , as you mention, to analyse more samples with season variation.
Author Response
Revision letter
Replies to each reviewer comment are provided bellow as RxRy (with x being the number of the reviewer and y the number of the comment by the same reviewer).
The anonymous reviewers are acknowledged, as their constructive criticism helped to improve the overall quality of the final manuscript.
Reviewer 4
R4C1: line 54 purposes
R4R1: Corrected as suggested.
R4C2: line 311 environmently safer please explain what that aims to there is no effect of the metod on profiles on the environment. I believe you want to explain the effect of the knowledge on the possibility to develop environmental work for the sustained populations. Please try to reformulate to improve the understanding.
R4R2: This sentence was revised to clarify any potential misunderstanding. It now reads as follows: “ The use of FA profiles of soft tissues and EF of mineral structures in marine species has been optimized to put forward faster and more accurate methods of analysis that can also reduce potential environmental impacts (associated with the residues they generate) that allow to best discriminate the geographic origin of these organisms, namely those that feature an important commercial value” (Lines 326-330).
R4C3: I suggest that you specify the season when catch of samples was performed at different locations, to be able to exclude season as the reason to your results..Temperature and light have effect on algal composition and thereby the food chain.
R4R3: To best accommodate this important suggestion by Reviewer 4, we have revised the “Sample collection” section in the “Material and methods”. It now reads as follows: “…during low tide during the Spring of 2015…” (Line 137)
R4C4: It is also needed, as you mention, to analyse more samples with season variation.
R4R4: The reviewer raises a very important question. Indeed, seasonal changes in the elemental and biochemical fingerprints of this species will be addressed in future works, whose sampling is already ongoing. It is paramount to understand how seasons affect the determination of geographic origin in order to apply this approach in real case scenarios, namely by authorities providing expert evidence for prosecution. It is crucial to know the geographic origin of some of the samples used to generate the predictive models that will try to allocate blind samples (origin unknown) to a given location (or classify them under an option termed “none of the locations surveyed”, to avoid bias promoted by over classification success forced by the statistical tools employed).
Reviewer 5 Report
Comments and Suggestions for Authors
This MS examines the potential of fatty acid profiles and elemental fingerprints as a tool to confirm the origin of an exploited polychate species and regulate its fisheries as bait. It is a very interesting subject; the applied methodology is well scheduled and conducted and the results are well presented and discussed. Therefore I am glad to suggest a publication of this work.
I only have some minor comments and corrections
1. The "simple summary" needs further simplification, and do not include the nominator of the species.
2. Please add some basal biological data for the species -if available-, especially on lifespan and reproduction in the study area and asses weather they conform to existing fisheries regulations (briefly).
3. Enlarge the photo of the species and possibly present it separately and add the copyright.
4. Include some biometrical information on the analyzed specimens in methods; it is important to know if they were at similar size/age
5. Why choosing Bray-Curtis in FA analysis and Euclidean distance in EF? please provide an explanation
6. The x-axis in Figure 2 is unreadable, please enlarge.
7. Enlarge the key in Figure 3.
8. Ln54 correct the word proposes with purposes, and improve the grammar of this sentence.
9. Ln153 correct the word bioamass with biomass.
10. Ln 168, 171 correct the chemical symbols with proper subscripts
11. Ln187, 192, 193, 202, 219, 220 delete parentheses
12. Ln289, 290 Delete the last sentence from the Table's legend.
13. Ln377, 382 replace shrimps with prawns when refer to Penaeidae
Comments on the Quality of English Language
English language is fine, only minor editing is needed.
Author Response
Revision letter
Replies to each reviewer comment are provided bellow as RxRy (with x being the number of the reviewer and y the number of the comment by the same reviewer).
The anonymous reviewers are acknowledged, as their constructive criticism helped to improve the overall quality of the final manuscript.
Reviewer 5
R5C1: The "simple summary" needs further simplification, and do not include the nominator of the species
R5R1: To best accommodate the positive criticism by Reviewer 5, we have revised the Simple Summary, which now reads as follows: “The overexploitation of the bristle worm Diopatra neapolitana in Ria de Aveiro, a coastal lagoon in mainland Portugal, has led to a generalized decline of its local populations, as it is commonly used as live bait for sports fishing. Several management actions have been put forward to reduce the impact of its harvesting, although illegal poaching stills threatens the sustainable use of this marine resource. In an attempt to verify if D. neapolitana was sourced from no take zones or if it was indeed collected from the place of origin claimed by live bait traders, this study evaluated if the geographic origin of D. neapolitana could be correctly assigned using a combination of fatty acid profiles and elemental fingerprints of its whole body and jaws, respectively. Results showed that both fatty acid profiles and elemental fingerprints differ significantly among locations, making it possible to discriminate the geographic origin of D. neapolitana. This discrimination achieves an even higher accuracy when combining these two natural barcodes than when employing each one of them individually. The present work can therefore contribute to the enforcement of management plans for the sustainable use of this commercially important marine resource.“
R5C2: Please add some basal biological data for the species -if available-, especially on lifespan and reproduction in the study area and asses weather they conform to existing fisheries regulations (briefly).
R5R2: This is an important suggestion by Reviewer 5, as indeed these topics have been overlooked in the original version of our manuscript. We have now added in the introduction information about the biological data of Diopatra neapolitana and it now reads as follows: “The reproductive biology of this species is relatively unknown [7]. D. neapolitana is a broadcast spawner with free-swimming larvae. In Ria de Aveiro, the main reproduction peak occurs from May to August and the male: female sex ratio is about 1:1 along the year [7]. This species has a regenerative capacity, being able to survive when a few anterior chaetigers are removed, mainly by predation. However, when D. neapolitana is harvested, usually, more than 20 chaetiger are harvested, compromising the survival of the posterior part of the specimen that remains in the burrow it inhabits [7].” (Lines 63-70).
R5C3: Enlarge the photo of the species and possibly present it separately and add the copyright.
R5R3: Corrected as suggested.
R5C4: Include some biometrical information on the analyzed specimens in methods; it is important to know if they were at similar size/age
R5R4: To best accommodate this important suggestion by Reviewer 5, we have now added information on the size of the specimens used in our study and it now reads as follows: “A total of forty adult specimens of D. neapolitana (total body length ranging from 314 mm to 584 mm) were randomly collected during low tide during the Spring of 2015 using a shovel, mimicking the method used by professional bait collectors in Espinheiro (E), Ílhavo (I) and Mira (M1 and M2) channels located in Ria de Aveiro, mainland Portugal (Figure 1).”. (Lines 136-140).
R5C5: Why choosing Bray-Curtis in FA analysis and Euclidean distance in EF? please provide an explanation
R5R5: This is a pertimnent question raised by Reviewer 5. We can use Bray-Curtis similarities or Euclidean distances in FA analysis, both approaches are correct. However, as our approach raised doubts in the reviewer and, it may also raise doubts on future readers of our manuscript, we have decided to reformulate our statistical analysis and solely use the Euclidean distance. This shift did not cause any significant changes on the results obtained as can now be seen in Table S2. As such, our main concluions remain valid.
R5C6: The x-axis in Figure 2 is unreadable, please enlarge.
R5R6: Corrected as suggested.
R5C7: Enlarge the key in Figure 3.
R5R7: Corrected as suggested.
R5C8: Ln54 correct the word proposes with purposes, and improve the grammar of this sentence.
R5R8: Corrected as suggested.
R5C9: Ln153 correct the word bioamass with biomass.
R5R9: Corrected as suggested.
R5C10: Ln 168, 171 correct the chemical symbols with proper subscripts
R5R10: Corrected as suggested.
R5C11: Ln187, 192, 193, 202, 219, 220 delete parentheses
R5R11: Corrected as suggested.
R5C12: Ln289, 290 Delete the last sentence from the Table's legend.
R5R12: Thank you for spotting this mistake. We have corrected the Table heading accordingly.
R5C13: Ln377, 382 replace shrimps with prawns when refer to Penaeidae
R5R13: Corrected as suggested.